# Adaptation, Feasibility, and Acceptability of a Health Insurance Literacy Intervention for Caregivers of Pediatric Cancer Patients (CHAT-C)

**DOI:** 10.3390/curroncol32020069

**Published:** 2025-01-28

**Authors:** Amy Chevrier, Perla L. Vaca Lopez, Katie Rogers, Monique Stefanou, Karely M. van Thiel Berghuijs, Douglas Fair, Elyse R. Park, Anne C. Kirchhoff, Echo L. Warner

**Affiliations:** 1Huntsman Cancer Institute, Salt Lake City, UT 84112, USA; amy.chevrier@hci.utah.edu (A.C.); perla.vacalopez@hci.utah.edu (P.L.V.L.); katie.rogers@hci.utah.edu (K.R.); monique.stefanou@hci.utah.edu (M.S.); karely.vanthiel@hci.utah.edu (K.M.v.T.B.); 2Department of Pediatrics, University of Utah, Salt Lake City, UT 84112, USA; douglas.fair@hsc.utah.edu; 3Primary Children’s Hospital, Salt Lake City, UT 84113, USA; 4Health Promotion & Resiliency Intervention Research Program, Mongan Institute, Boston, MA 02114, USA; epark@mgh.harvard.edu; 5Departments of Psychiatry & Medicine, Mass General Hospital, Harvard Medical School, Boston, MA 02114, USA; 6College of Nursing, University of Utah, Salt Lake City, UT 84112, USA

**Keywords:** caregiver, cancer caregiver, pediatric cancer, health insurance literacy, financial hardship, financial toxicity

## Abstract

We adapted CHAT, a four-session virtual program to help individuals affected by cancer manage insurance and medical costs for caregivers of pediatric cancer patients (called CHAT-C); we then pilot-tested CHAT-C. Eligible caregivers were ages 18+ and the primary caregiver to a pediatric cancer patient (≤25 years old) diagnosed in the past five years and treated at Primary Children’s Hospital. We conducted engagement studios to adapt the program. Feedback was evaluated using a rapid qualitative analysis framework and included content preferences, navigator preferences, logistics/structure, timing of delivery, and feasibility/acceptability. A small pilot test of CHAT-C was conducted; feasibility, acceptability, and preliminary efficacy were evaluated based on enrollment rates, qualitative feedback, and baseline/follow-up surveys. Participants in the pilot (*n* = 14) were primarily white (93%), married (93%), female (86%), ages 40–49 (50%), and college-educated (57%). Most participants (64%) completed all four sessions of CHAT-C. Those who did not complete the sessions cited a lack of time, a child’s disease progression, and a perceived lack of benefit. Health insurance literacy (measured by nine items) improved by 10.8 points on average (SD = 6.0, range: 9–36) after CHAT-C. Caregivers of childhood cancer patients are willing to participate in a health insurance program, but some caregivers need less time-intensive options.

## 1. Introduction

Caregivers of pediatric cancer patients experience high levels of stress [1,2]. Along with the psychological stress of having a child undergoing cancer treatment, caregivers experience increased responsibilities as they seek to manage caring for the emotional and physical needs of their ill child, support the needs of other children, moderate their own emotions, navigate their child’s care, adapt family routines to accommodate their child’s treatment, and balance work and other demands of daily life [1]. The intensity of their child’s treatment, marital strain, lack of social support, and financial strain can all negatively impact caregivers of pediatric cancer patients [1,3]. Many caregivers of pediatric cancer patients may experience unique challenges due to their life stage–caregivers of pediatric cancer patients are, on average, 35.7 years old [4,5]. Young adult caregivers report higher levels of stress, which is likely due to other developmental tasks typical in this stage of life [5]. Caregiving as a young adult can interfere with obtaining an education, maintaining a social network, finding a spouse, and starting a family [5]. Young adults have not achieved the financial and employment stability that older caregivers often have and may experience increased financial stress as well as physical and financial stress from having to maintain a job while providing care [5]. Since many caregivers of pediatric cancer patients are young adults, this population may be at higher risk for poor psychological and financial outcomes.

Finances can be a significant source of caregiver stress [6]. Cancer treatment oftentimes leads to significant out-of-pocket expenses, which can result in negative impacts on health-related quality of life (financial toxicity), and caregivers of younger patients have been shown to be particularly at risk [7]. Changes to employment that occur as a result of a cancer diagnosis in the family and the associated changes to insurance coverage may place the family’s financial stability at risk [7]. Due to the financial risks, many caregivers are reluctant to make changes to their employment even while navigating the responsibilities of caring for a cancer patient [8]. Thus, caregivers may work reduced hours, exhaust paid time off, take unpaid leave, or even increase their work hours due to the costs of cancer care [8].

Difficulties in navigating the healthcare system can be a significant source of stress for caregivers and can result in poor mental and physical health outcomes for the caregiver as well as the patient [9]. Interventions exist to support cancer survivors in navigating the financial impacts of a cancer diagnosis [10], but to our knowledge, none of these interventions have been created or modified for cancer caregivers of pediatric patients. Survivor-focused interventions for improving health insurance literacy, as well as the knowledge and application of health insurance concepts, have demonstrated efficacy that extends beyond insurance literacy and could potentially improve financial toxicity [11,12]. Adequate health insurance literacy is a necessary skill to manage healthcare costs and obtain appropriate cancer-related care [13]. Many cancer patients and their caregivers, however, report a need for improving both financial and health insurance literacy [14,15]. Therefore, to best support cancer survivors and families as a unit, health insurance education interventions need to be adapted for caregivers of cancer patients. Additionally, helping caregivers improve their health insurance and financial literacy could help them feel more confident and more in control of their financial situation. Most significantly, ensuring that caregivers know how to navigate insurance and costs from the beginning of the cancer trajectory is critical to support the transition into survivorship so that pediatric cancer survivors have seamless follow-up care to identify and manage any future late effects.

To address these gaps, we adapted CHAT (“Let’s CHAT about health insurance”) [11], our previously tested virtual health insurance navigator-delivered education program focused on adolescent and young adult cancer patients, for the caregivers of pediatric cancer patients (CHAT-C). The CHAT program includes four educational sessions with a patient navigator and an educational booklet covering the content of the program. Topics of the sessions include understanding insurance terminology and structure, reviewing their insurance plans, situations where out-of-pocket costs arise, tips on budgeting, and how to manage conversations around costs with providers. Our goal is to design a program to improve health insurance literacy that can empower caregivers to make informed decisions about their health insurance, better advocate for their children, and feel more in control of their finances. Additionally, enhancing caregivers’ understanding of health insurance options and legislative protections may reduce their financial toxicity and support their child’s long-term health. This paper focuses on the initial adaptation of CHAT-C through community feedback and pilot testing of the CHAT-C program. The purpose of this research was to obtain specific feedback regarding the content and delivery of the CHAT-C program and pilot test the program for feasibility, acceptability, and preliminary efficacy.

## 2. Materials and Methods

We implemented a mixed-methods study, including gathering feedback from the target population and pilot-testing CHAT-C with caregivers of pediatric cancer patients. Prior to this step, we altered the original patient examples of CHAT to include a pediatric cancer patient and tailored the language to be directed at caregivers. To gather community feedback on CHAT-C, we partnered with the Clinical and Translational Science Institute (CTSI) Community Engagement and Collaboration Team at the University of Utah to conduct two engagement studios [16] with caregivers (Aim 1). Concurrently, we pilot-tested the CHAT-C program (Aim 2). Our overall goal was to evaluate the feasibility, acceptability, and preliminary efficacy of CHAT-C for improving health insurance literacy among cancer caregivers for later testing in a larger, randomized trial. Ethics approval was provided by the University of Utah, IRB_00163219.

### 2.1. Recruitment and Participants

All participants were recruited from Primary Children’s Hospital (PCH) in Salt Lake City, UT, from May 2023 to December 2023. PCH treats between 220–240 new pediatric cancer diagnoses each year. For both aims of the study, eligible participants were adult (18+ years old) caregivers (typically a parent) of cancer patients who were diagnosed between ages 0 to 25 and treated within the past five years at PCH. Participants also needed to speak English and have access to a smartphone, tablet, or other video-capable device. The participants were recruited by email, text message, and phone call. Identification of the eligible caregivers for both aims occurred by screening clinic schedules and physician referrals from PCH.

For Aim 1, those interested in participating in the engagement studios enrolled by completing a short online survey through REDCap that included demographic questions and a consent cover letter. These individuals were then contacted by the CTSI team to determine the best time to schedule the session. Actual participation was based on availability. Upon completion of the engagement session, the participants received a USD 75 gift card. Due to the focus on community feedback for this aim and to reduce participant burden, we collected limited demographic information on the participants.

For Aim 2, the pilot intervention, the participants enrolled by completing a baseline survey, including a consent cover letter and questions regarding demographic information and health insurance literacy/financial stability. All participants were given the opportunity to participate in the educational program. A follow-up survey was sent one week after the program’s completion (those participants who opted out of the sessions were sent their follow-up survey after indicating they no longer wanted to participate in the sessions). The participants received a USD 25 gift card for each completed survey.

### 2.2. Data Collection—Engagement Studios

For Aim 1, engagement studios occurred in June and August 2023. Engagement studios are two-hour meetings with the participants who are considered “community experts” based on their specific lived experiences. We have successfully used engagement studios for earlier projects, including the adaption of a Spanish version of CHAT for individuals without cancer. The engagement studios focused on obtaining caregiver feedback on the educational program, including potential adaptations specific to caregivers and the program structure, timing, and delivery. Before the engagement studios, the participants were emailed a PowerPoint presentation overview of the program. They were also sent the list of questions that would be addressed in the engagement studio. A representative of the study team presented the same PowerPoint presentation at the beginning of the engagement studio. The presentation included the goals and structure of the program, a high-level overview of the content of the program, and a few examples of the graphics from the educational booklet. After presenting, the study staff remained on the call but did not participate unless asked to by the CTSI moderator. The moderator led the discussion to limit the study staff’s influence on the feedback obtained from the participants. The list of prepared questions asked by the moderator explored the caregivers’ (1) preferences for the content and topics of CHAT-C, particularly as they related to caregivers of pediatric cancer patients; (2) caregiver ideas for improving understanding of the materials and engagement in CHAT-C; and (3) caregiver preferences for the delivery of CHAT-C, including recommendations about the number of sessions, time spent for each session, and who should attend the sessions.

### 2.3. Data Collection—Pilot

The goal of Aim 2 was to pilot-test CHAT-C with the caregivers to refine the intervention by testing the feasibility of the program with the target population, the acceptability of the content, the potential implementation needs, areas of improvement, and information on preliminary efficacy for improvements in health insurance literacy. The CHAT-C intervention included four 30–45 min one-on-one educational sessions with a patient navigator, occurring virtually over Zoom 6.3 (San Jose, CA, USA) over the course of approximately four weeks (scheduling was flexible depending on the participant’s availability). More information on the program content is available elsewhere [17]. The sessions were audio recorded and transcribed.

We evaluated the feasibility, acceptability, and preliminary efficacy of our approach. All participants completed a brief survey that included health insurance literacy and financial toxicity questions [18,19], as well as familiarity with the Affordable Care Act (ACA) protections/provisions, such as dependent coverage, employment status, insurance status, as well as policyholder, family relationships, and sociodemographics [11]. The follow-up survey included the same health insurance literacy and financial toxicity questions and also the Patient Satisfaction with Navigator Interpersonal Relationship (PSN-I) [20]. Given that CHAT-C is delivered virtually over Zoom, confidence in technology use was evaluated for feasibility [21]. The follow-up survey also included an open-ended question to obtain feedback on the acceptability of the content and potential implementation issues.

### 2.4. Data Analysis

We applied descriptive statistics to summarize the sociodemographic- and cancer-related factors for the caregivers and the pediatric cancer patients in their care. Feasibility was determined by evaluating the proportion of caregivers enrolled out of those approached, the number of sessions each caregiver completed out of the four total sessions, and the participants’ confidence in using technology [20]. Acceptability was determined through feedback from open-ended questions in the follow-up survey for the pilot about the content and potential implementation issue areas and through the PSN-I score, which is a 9-item measure of patient satisfaction with their navigator’s interpersonal skills, technical competence, and accessibility [20].

Preliminary efficacy was determined by comparing health insurance literacy, financial toxicity, and ACA familiarity from the baseline to follow-up using paired *t*-tests. Health insurance literacy was evaluated using selected items from the Health Insurance Literacy Measure (HILM), which assesses a participant’s confidence in navigating the health insurance system; 9 items from this measure were included and scored, ranging from 9–36, with higher scores signifying higher insurance literacy [19]. Financial toxicity was measured as the 11-item COmprehensive Score for financial Toxicity (COST), with a score ranging from 0–44, with lower COST scores indicating greater financial hardship [18]. ACA familiarity was calculated as the sum of items asking about familiarity with ACA protections/provisions, with a score ranging from 0–6, with higher scores indicating higher knowledge of the ACA; we also report the individual ACA items. An α level of 0.05 was used to determine the statistical significance. All statistical analyses were performed in Stata 18.0 (College Station, TX, USA).

### 2.5. Data Integration

To enrich the interpretations of feasibility, acceptability, and preliminary efficacy, we integrated qualitative feedback from the engagement studios via rapid qualitative analysis [22]. The categories, strength (weak or strong influence on adaptation), and valence (positive or negative influence on adaptation) of feedback were documented during the engagement studios by a member of the CTSI team. Then, two authors, AC and PVL, reviewed the notes from the engagement studios with the full transcripts to identify any potential discrepancies in the emergent categories, strength, and valence of feedback. Attention was paid to the emergent codes that were relevant to the adaptation of CHAT-C. Comments and highlighting were used to identify discrepant sections of the transcribed engagement studios for discussion with a third member of the study team, ELW, for adjudication. The analytic team met twice to review, adjudicate differences, and refine the coding. Following discussions, the primary analysts (AC and PVL) wrote a high-level summary for each content area, which was reviewed and edited for clarity about the valence and strength of the feedback. Valence and strength are indicated in the results as the extent to which the majority or minority of participants endorsed certain points of feedback. Finally, a joint display was created to map agreement and discordance in satisfaction.

## 3. Results

### 3.1. Caregiver Sociodemographics and Patient Cancer Factors

Of the 28 participants enrolled to participate in the engagement studios, only *N* = 12 were able to participate due to scheduling conflicts (*n* = 6 per engagement studio; Table 1). The engagement studio participants’ age range was 31–56 years (50% were between 40–49 years of age at the time of the engagement studio), 91.7% were white, female, and lived in an urban/suburban area. The majority, 58.3%, reported a total annual income of ≥ USD 75,000.

For the pilot intervention, *N* = 14 caregivers consented. These caregivers were primarily white (92.9%), married (92.9%), female (85.7%), college-educated (57.1%), and ages 40–49 (50.0%). Most pediatric cancer patients were off treatment (78.6%). At baseline, most patients (85.7%) had health insurance under their parent’s policy; 71.4% of caregivers reported that their spouse/partner was the policyholder for their child’s insurance. Patients’ ages at diagnosis ranged from 4–15 years old (mean: 8.75; median: 8). Patients were, on average, 1.8 years from diagnosis (median: 1.6; range: 0.5–4.4 years).

### 3.2. Engagement Studio Feedback on the Adaptation of the CHAT for Caregivers

Feedback from the engagement studios emphasized five areas of change for adapting the CHAT-C program, including content preferences, navigator preferences, logistics/structure, the timing of delivery of the program, improving audience motivation, and feasibility/acceptability. These subcategories of feedback are described below, with supportive quotes in Table 2.

#### 3.2.1. Content Preferences

Participants in the engagement studios offered many suggestions for possible content changes to CHAT-C to make it relevant for caregivers. The most recurring feedback was tailoring the information about health insurance to caregivers’ baseline knowledge and familiarity with insurance. They recommended specific content, including information about healthcare laws and parents’ rights, explanations of different types of health insurance, an overview of the explanation of benefits in contrast to bills, information on billing insurance, information on deductibles, having caregivers bring their own explanation of benefits and insurance card to discuss, and providing caregivers with something to refer back to review the content of the sessions.

Caregivers also desired a list of helpful contacts, such as social workers and health insurance contacts. Caregivers wanted more specific lists of who to contact for different billing/insurance issues, more specific information on the functioning of the billing department at Primary Children’s Hospital (including who to contact for specific issues) and explanations of billing codes, examples of questions to ask their insurance company, and information on paying for treatment-related medical devices, such as wigs and wheelchairs. Another highly recommended change was adding more content specific to Medicaid, as the original CHAT program contains more of a focus on commercial insurance plans. Less common content suggestions included adding an introduction page for the person delivering the program, a section on taxes, instructions on how to store and organize bills and other important documents, information on navigating secondary insurance, content on life insurance, and strategies for obtaining the most out of a flexible spending account or a health savings account.

#### 3.2.2. Navigator Preferences

The participants had few preferences about the specific role of the person delivering CHAT-C as long as they could teach in an accessible and sensitive way, were empathetic and understanding of the caregivers’ situation, and were capable of framing the content to make participants feel empowered rather than placing blame. The participants emphasized that the person delivering the sessions should not be an employee of insurance companies, as they perceived that the navigator should be their advocate and not an agent for the hospital or insurance company. Other suggestions from caregivers focused on ensuring that the person delivering the sessions was knowledgeable of the caregivers’ specific situation to facilitate tailoring of the content. The caregivers also want a navigator who is respectful of questions and gaps in understanding, knowledgeable about the specifics of insurance, and quick to answer questions, even if that means quickly finding answers to questions that might be unknown at the time of the session.

#### 3.2.3. Logistics/Structure

The most repeated feedback on the structure of the program was to tailor the information covered and the number of sessions to each individual participant—including making the delivery of the sessions non-linear to let participants go through the subjects in the order they preferred. One suggestion was to divide the sessions into basic, intermediate, and advanced modules. Many suggested implementing assessment and teach-back strategies to solidify learning but avoiding gamification of the program—emphasizing that the information should not feel like it is being taken lightly.

Participants had a strong preference for live program delivery rather than prerecorded videos but were strongly in favor of having the option to record the sessions for future reference. It was suggested that in-person and virtual delivery options should be available. While some participants felt like four sessions was too many, others felt that more sessions that were shorter and covered less information would be preferable. It was largely agreed upon that participants would benefit from a follow-up review session either weeks or even months after completing the program.

Other suggestions included creating a workbook to be filled out during the sessions to promote engagement. Other feedback regarding the format of the program included sending the vocabulary words in advance to save time, renaming sessions to have more interesting names, moving the “how to read a bill” section to the last session that is focused on finances, and to separate a session that included both understanding your specific insurance policy and how to read a bill into two sessions or extending the delivery time due to the complexity of the content. They also suggested that participants should bring their own bill to interpret rather than using an example bill.

#### 3.2.4. Timing of Delivery

The timing of the program was not as unanimously agreed upon, except that it should be flexible. The participants were split on starting the program when treatment was beginning, which can be a very overwhelming time period, while others thought it best to have the program offered to caregivers as soon as possible so they can be aware of their insurance options and rights. Knowledge of CHAT-C as an available resource earlier in the diagnosis trajectory was seen as a tool to enable caregivers to make more informed decisions about handling bills and choosing insurance plans.

For those who wanted health insurance education at the beginning of treatment, recommendations included advertising the program with a handout in the initial treatment information packet or on clinic bulletin boards. Caregivers emphasized the importance of advertising so that the program could be completed on their own timeline. Another suggestion that was supported by most participants was to time the program’s advertising close to the open enrollment periods for insurance or when the first treatment bills would be due. The participants who wanted health insurance education after initial treatment suggested that caregivers could be invited to take part in the program regularly throughout treatment and survivorship—as interest in the program may change with the caregiver’s current situation.

#### 3.2.5. Audience

The participants felt strongly motivated to participate in CHAT-C, as they perceived the information to be very important to any caregiver with a child receiving cancer treatment. The participants also said that it would be ideal to have spouses/partners join in with the program or possibly another family member. They also suggested that it could be helpful to have a healthcare team member, such as a social worker, join as well. Representatives from insurance companies and human resources also emerged as key stakeholders to be involved in CHAT-C’s delivery.

#### 3.2.6. Feasibility/Acceptability

All participants said that the CHAT-C content seemed useful, relevant, and acceptable. However, some participants raised concerns about completing the sessions due to limited time. The participants felt that high-level summaries of health insurance information (e.g., avoiding long paragraphs of text) would be acceptable to reduce the time required to complete the sessions. Even participants who were knowledgeable about health insurance felt that CHAT-C could be beneficial due to the complexity of the insurance coverage structure.

### 3.3. Pilot Intervention

#### 3.3.1. Feasibility

There were *n* = 39 caregivers approached for enrollment in the CHAT-C pilot intervention sessions, of which *n* = 11 enrolled. Of the other 28, *n* = 20 did not respond to contacts, *n* = 6 declined (*n* = 3 were not interested, *n* = 2 could not participate due to lack of time, *n* = 1 already had a navigator helping them with insurance), and *n* = 2 were found to be ineligible due to language barriers. All participants in the engagement studios were also offered the opportunity to take part in the CHAT-C pilot. Of the *n* = 4 who indicated an interest in participating in the engagement studios, *n* = 3 enrolled, and *n* = 1 later declined due to the intensity of their child’s treatment. In total, *N* = 14 caregivers enrolled in the CHAT-C pilot, for an overall participation rate of 34.1% (14/41 eligible).

Of the 14 participants, at baseline, the majority were confident in using technology-based mobile applications without help (92.9%), felt confident in setting up a video chat without help (71.4%), and believed they could solve basic technology problems on their own (78.6%).

A total of *n* = 9 caregivers completed all four sessions (64.2%), and one completed three sessions but did not complete session 4. Four caregivers completed zero sessions (*n* = 2 child became too ill/caregiving was too intense to participate, *n* = 1 was lost to follow-up, and *n* = 1 declined because they did not feel they would benefit from the program. On average, the sessions lasted 35 min each (range 24–55 min). Overall, 11 caregivers completed the follow-up survey (10 who did at least one session and 1 who did not do sessions).

#### 3.3.2. Acceptability and Satisfaction

Of those who completed the sessions (*n* = 10), all reported that they would recommend the program to other caregivers. Of the *n* = 6 participants who offered write-in feedback, *n* = 4 said they would make no changes to the program, *n* = 1 said they would like weekend availability for the sessions, and *n* = 1 wanted more details about what the program entailed prior to enrollment. Satisfaction with CHAT-C was high, with over 90% of the participants reporting they were very or somewhat satisfied with the overall quality of the program, working with the insurance navigator, the ability to schedule sessions at a convenient time, the number and length of the sessions, the accompanying health insurance handbook, and the use of video conferencing (i.e., Zoom) (Table 3). All participants indicated they would recommend the program. The participants also said they were looking forward to sharing the knowledge they acquired in CHAT-C with others.

#### 3.3.3. Preliminary Efficacy

At baseline, for the 14 participants, the mean health insurance literacy and COST scores were 23.6 (SD = 7.1) and 20.3 (SD = 10.8), respectively (Table 4). Knowledge of the Affordable Care Act provisions was 2.7 (SD = 2.3). For the individual ACA items, dependent coverage for insurance was the item with the highest familiarity (78.6%), with several other provisions indicated by 50% or fewer of participants (appeals, pre-existing conditions, etc.), and the familiarity that insurers can no longer set annual/lifetime dollar limits was lowest (14.3%).

Among the participants who completed the CHAT-C sessions (*n* = 10), the health insurance literacy scores improved from baseline to follow-up by an average of 10.8 points (SD = 6.0; *p* < 0.01; baseline mean = 22.8 (SD = 7.5) and a follow-up mean = 34.1 (SD = 3.3)). The COST scores did not change (mean change = 0.5; SD = 8.6; *p* = 0.87; baseline mean = 22.8 (SD = 9.4) and a follow-up mean = 24.7 (SD = 8.8)). ACA familiarity improved by a mean score of 2.9 (SD = 2.4; *p* < 0.01; baseline mean = 2.9 (SD = 2.2) and a follow-up mean = 5.8 (SD = 0.4), Table 4).

## 4. Discussion

Due to the complexity of health insurance coverage in the United States, families affected by cancer often face unmet needs regarding managing care and healthcare costs that require support and education [23]. In this study of the adaptation and initial pilot testing of the CHAT-C intervention, which is a patient navigator-delivered health insurance literacy program for caregivers of childhood cancer patients, we found that most caregivers feel that the program is needed, feasible, and acceptable; several mentioned a need to tailor the content to specific knowledge levels. The caregivers who participated in the pilot showed improvements in health insurance literacy and knowledge of the Affordable Care Act provisions. Lack of time for the intervention was raised as the primary barrier to CHAT-C participation, given the caregivers’ many competing demands. Thus, by coupling the feedback from the caregivers through both engagement studio feedback and an initial pilot, CHAT-C can undergo further refinement prior to a larger implementation, including identifying flexible strategies that incorporate existing knowledge levels on insurance for participation by busy caregivers.

The participants in the engagement studios overall endorsed the CHAT-C content by expressing that it was relevant to their needs and circumstances; similarly, in the pilot trial, the participants were, overall, satisfied with the program’s content. Many suggestions that were brought up in the engagement studios were already included in the program, such as information on insurance-related laws, indicating the acceptability of the content. However, multiple participants indicated a need to tailor the information to caregivers’ knowledge of the healthcare system and insurance, which can vary greatly depending on their personal experiences and education on the topic [24]. For example, when we examined the markers of insurance knowledge at baseline, 78.6%% of the sample knew about key ACA protections, such as dependent coverage until age 26, whereas very few (14.3%) knew that insurance companies could no longer set annual limits on coverage—both major factors that potentially protect consumers financially.

Going forward, efforts to tailor CHAT-C’s information should ensure that key knowledge gaps are still addressed for all participants while also allowing them to indicate areas of navigation preference. For example, future iterations of CHAT-C could include having participants fill out a short survey prior to the first session to choose which topics they want to cover, as well as choices regarding the length and number of sessions. However, caregivers may not always be aware of what they do or do not understand regarding insurance, which could pose an issue for implementing this approach [25]. Thus, a SMART design trial allowing for intervention adaption based on participant responses could be an important consideration for future implementation [26]. Additionally, choices of allotted time for session delivery would also have to be dependent upon the number of topics that caregivers would like to or need to have covered, which may be difficult to execute appropriately due to navigator capacity.

Of the participants approached for the pilot, 34.1% enrolled, which is not unlike the recruitment rates for other studies recruiting cancer caregivers [27,28]. In the engagement studios, we heard that caregiver time restrictions often emerged due to changes in the current health status of their child and demands on the caregiver due to their child’s treatment. These demands may vary from day to day due to the dynamic nature of cancer treatment. The current CHAT-C protocol allows caregivers who are unable to participate when approached to reach back out to the research team when they have more availability, but guaranteeing that these caregivers have the opportunity to participate might require a more systematic follow-up over time.

### 4.1. Implications

While the participants did not like the idea of prerecorded videos as a form of program delivery, this preference was largely based on wanting the opportunity to ask questions to the navigator. The participants did endorse sending materials before the sessions to review prior to meeting with the navigator. This suggests that participants may be open to prerecorded material, such as videos, as long as they are able to still meet live with and feel supported by a navigator. As a mode of delivery, prerecorded content could be divided into easy, intermediate, and advanced modules (as suggested in the engagement studios) with set time lengths, which may be the most efficient way to allow busy caregivers to participate. Giving caregivers the opportunity to review certain material on their own timeline while still having the opportunity to ask questions and feel supported by a navigator would allow for better tailoring of the topics and a more flexible schedule for covering the material—both of the most often repeated suggestions during the engagement studios. Furthermore, from a scalability standpoint, integrating videos that deliver some of the session content could also help with ensuring more caregivers are able to receive the CHAT-C program content.

Thus, future steps in this research include further refinement of CHAT-C by implementing the feedback gathered in this study, including allowing for scheduling flexibility or shorter sessions due to busy schedules. Additionally, we plan to gather more feedback from caregivers on content delivery (e.g., short videos) as well as the dose of content needed, which is particularly important as we integrate the suggestions for tailoring the material.

### 4.2. Limitations

Our study had several limitations that may impact the findings. As this was focused on adaptation and initial pilot testing, we had a small sample, which limits the range of experiences and perspectives and may not be representative of the experience and perspectives of all caregivers. The recruitment process took place in one pediatric hospital located in the Salt Lake Metropolitan area, which serves individuals from the Five-State Intermountain West. Most of our sample were married, middle-aged, suburban, white, middle-class women with a limited representation of diverse gender identities, racial or ethnic backgrounds, age groups, marital statuses, rural/urban living situations, and socioeconomic statuses, which could influence caregiver experiences. Caregivers from different identities and other regions may have distinct needs and experiences when caregiving for a pediatric cancer patient, which was not highlighted in these findings. Additionally, CHAT and its sister study, HINT, were designed for young cancer patient and survivor populations. While both programs have shown improvements in health insurance literacy in pilot trials [11,12], ongoing larger trials are currently being run to ascertain longer-term impacts on outcomes such as surveillance care for recurrence and access to survivorship care. Thus, CHAT-C’s potential for affecting patient health outcomes is unknown and will be a focus for future work. Finally, while most participants reported high levels of comfort using technology, study activities were completed online, and those who were uncomfortable using technology would likely not choose to participate.

## 5. Conclusions

In summary, in this study of the initial adaptation and testing of the CHAT-C program for caregivers of pediatric cancer patients, we found that caregivers are willing to participate in a health insurance education program, but some caregivers need less time-intensive options. Subsequent work should also include testing of the program on a larger scale and expansion of the adaptation of the program to include caregivers from more diverse communities with a larger range of sociodemographic experiences, thus supporting all caregivers of pediatric cancer patients.

## Figures and Tables

**Table 1 curroncol-32-00069-t001:** Caregiver sociodemographics and patient cancer factors.

	Engagement Studios (*N* = 12)		CHAT-C Pilot Intervention (*N* = 14)	
	** *N* **	**%**	** *N* **	**%**
**Caregiver Ethnicity**				
Not Hispanic/Latino	#	#	12	85.7
Hispanic/Latino	#	#	2	14.3
**Caregiver Race**				
White	11	91.7	13	92.9
American Indian/Alaska Native	0	0	1	7.1
Prefer not to answer	1	8.3	0	0
**Caregiver Marital Status**				
Married	#	#	13	92.9
Divorced	#	#	1	7.1
**Caregiver Gender ***				
Female	11	91.7	12	85.7
Male	1	8.3	1	7.1
**Caregiver Education**				
College graduate (four-year degree) or higher	7	58.3	8	57.1
Some college/Associates/technical school	4	33.3	3	21.4
High school diploma/GED or less	0	0	3	21.4
**Caregiver age**				
18–29	0	0	1	7.1
30–39	4	33.3	4	28.6
40–49	6	50.0	7	50.0
50–59	2	16.7	2	14.3
**Child’s Insurance Type ****				
Insured through parent or guardian’s policy	#	#	12	85.7
Insured through Medicaid or other state program	#	#	5	35.7
**Child’s Insurance Policy Holder**				
Participant	#	#	3	21.4
Child	#	#	2	14.3
Participants’ Spouse/Partner	#	#	10	71.4
**Household Income**				
$20,000–39,999	#	#	1	7.1
$50,000–74,999	4	33.3	#	#
$60,000–79,999	#	#	5	35.7
$75,000+	7	58.3	#	#
$80,000–99,999	#	#	2	14.3
$100,000+	#	#	6	42.9
**Number of individuals in household ***				
3	1	8.3	1	7.1
4	3	25.0	4	28.6
5	3	25.0	7	50.0
6	3	25.0	2	14.3
7	1	8.3	0	0
**Rurality**				
Suburban	8	66.7	#	#
Urban	2	16.7	#	#
Rural	1	8.3	#	#
**Child’s treatment status**				
On treatment	#	#	3	21.4
Off treatment	#	#	11	78.6
**Child’s age at diagnosis**				
1–5 years old	#	#	6	42.8
6–10 years old	#	#	4	28.6
11–15 years old	#	#	2	14.3
16–20 years old	#	#	2	14.3

* Gender missing for (*n* = 1), number of individuals in household (1), income (1); # Questions not asked due to the limited factors ascertained on caregiver demographics for the engagement studios; ** Multiple options could be selected, so totals do not equal 100%.

**Table 2 curroncol-32-00069-t002:** Engagement studio participant feedback and supportive quotes for CHAT-C.

Subcategory	Participant Feedback	Supportive Quote
Content	Main recommendations included tailoring the program to each individual and for the program to include a contact list for all the different resources available, such as social work and financial advisors.	“Like maybe somebody is familiar with the first session and the information that they go over, can they opt out of the first two sessions and ask that the last two are accelerated and start there? I think people would be much more likely to participate in something that they don’t already have the knowledge of.” “I just want to say that [the] list of contacts and what they can help with is so valuable.”
Navigator	Participants did not express a preference for who should deliver the program but rather emphasized the qualities an individual should have, such as empathy, sensitivity, and expertise in the content presented.	“I would, again, raise it as an advocate program for the caregiver and be careful with that wording and that verbiage and what you start off with so that it appeals to the caregiver. And you can get that message across that this is here to help you. And we know you’re overwhelmed. And I think you really need to be careful in that space when parents or caregivers are dealing with that diagnosis.” “To have someone say I hear you and I’m here for you, let me help you would be the best thing.”
Logistics/structure	Participants requested for the program structure to be tailored to allow them to receive information in the order of their preference. They also expressed a preference for live sessions, enabling them to ask questions and receive immediate feedback.	“You might consider making it a non-linear format. By that I mean here’s your topics. Where do you want to start? Let the participant have a little more control over that.” “Like the delivery of the program, I definitely would like it live every time. I think it would just get our questions out and be able to have them answered. But also, I’m the type of person that really likes hard copies of things. So, having an option for a printable handout that I can take notes on and follow along and read, I think that would make me personally stay more engaged and focus more.”
Timing	Participants suggested that a flexible timeline for the program delivery would be ideal, considering the changing circumstances caregivers face.	“If somebody had come into our hospital room and said, “Do you want to talk about insurance and go to this little thing to teach you how to navigate your insurance?” Those first few weeks, I would have said, “Please leave me alone.” There would have been no room, no space for me to have that conversation.” “I think if maybe at the very beginning or the first week [of treatment], we could have been given a packet just saying, ‘Hey, this is available for you if you need it and when you’re ready.’”
Audience/Motivation	Participants expressed that the topic and information were highly important and felt no need for additional motivation. However, many suggested that having the option for a partner or spouse to join would be beneficial.	“This is so critically important that the chances that I’m going to fall asleep are probably if I’ve been up all night, not if I’m bored with the topic. Whether I can process it is another issue. But I don’t think it needs…Some optimistic and encouraging through process can be helpful for somebody getting through this. But it’s not a game. And I don’t know that I’d feel comfortable if it was treated too lightly.” “I’m sure all of us would want somebody else there to ask any questions that we miss.”

**Table 3 curroncol-32-00069-t003:** Satisfaction of CHAT-C program and navigation among pediatric cancer caregivers at follow-up (*N* = 10) *.

	Very/Somewhat Satisfied		Neither		Very/Somewhat Dissatisfied		Example Quote
	*N*	%	*N*	%	*N*	%	
Overall quality of program	10	100	0	0	0	0.0	“It was perfect” “Give a better explanation about what the sessions are about before. I had no idea I was going to be learning about insurance.”
Working with the insurance navigator	10	100	0	0	0	0.0	“[Navigator] was fantastic. She is personable, reliable, and cordial.” “[Navigator] was amazing and thorough! She is fantastic!” “[Navigator] was fantastic at helping me understand what I didn’t.”
Ability to schedule sessions at a convenient time	10	100	0	0	0	0.0	“Availability for weekends.”
Number of sessions	9	90	0	0	1	10	No comments
Length of sessions	9	90	1	10	0	0.0	No comments
Health insurance handbook	10	100	0	0	0	0.0	“I would just have the list of resources for families listed and available in the first session. That is what many families are in need.”
Zoom platform for sessions	10	100	0	0	0	0.0	No comments
Would you recommend the program	10	100	0	0	0	0.0	“Overall, this is a great tool to have.”

* Satisfaction reported for the 10 participants who participated in the sessions.

**Table 4 curroncol-32-00069-t004:** Baseline and follow-up health insurance literacy, COST, and familiarity with the ACA measures.

**Baseline survey (*N* = 14)**				
	Mean		SD	
Health insurance literacy	23.6		7.1	
COST *	20.3		10.8	
ACA familiarity *	2.8		2.0	
	**Yes**		**No/Don’t know**	
**ACA provision familiarity ***	** *N* **	**%**	** *N* **	**%**
Annual/lifetime limits	2	14.3	12	85.7
Dependent coverage to age 26	11	78.6	3	21.4
Free preventive care	6	42.9	8	57.1
File appeal	7	50.0	7	50.0
Pre-existing conditions	7	50.0	7	50.0
Subsidies for low-moderate income	6	42.9	8	57.1
**Follow-up survey (*N* = 10)**				
	**Baseline** **Mean (SD)**	**Follow-up Mean (SD)**	***p*-value**	
Health insurance literacy	22.8 (7.5)	34.1 (3.3)	<0.01	
COST *	22.8 (9.4)	24.7 (8.8)	0.87	
ACA familiarity *	2.9 (2.2)	5.8 (0.4)	<0.01	

* COST = COmprehensive Score for financial Toxicity; ACA = Affordable Care Act. Items shown in the mean score are the summary of the 6 individual items. SD = standard deviation.

## Data Availability

The raw data supporting the conclusions of this article are available from the authors upon request.

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
