# Peer review of "Adaptation, Feasibility, and Acceptability of a Health Insurance Literacy Intervention for Caregivers of Pediatric Cancer Patients (CHAT-C)"

_curroncol, 2025, doi:10.3390/curroncol32020069_

Round 1

Reviewer 1 Report

Comments and Suggestions for Authors

This article explores an area that is not often found in journals.  It would be interesting to see more studies with a larger sample size and more diverse population, especially those from an economically-disadvantaged background who would benefit the most from understanding how to navigate the complicated health insurance system.

Author Response

Thank you so much for taking the time to review this manuscript. I have addressed your comments below. 

Comment 1:

This article explores an area that is not often found in journals.  It would be interesting to see more studies with a larger sample size and more diverse population, especially those from an economically-disadvantaged background who would benefit the most from understanding how to navigate the complicated health insurance system.

Response 1:

Thank you for your feedback. We agree that diversifying our population for future research would be a good future step; we have added a note on page 12 of the manuscript to clarify that we plan on testing on a larger scale. “Subsequent work should also include testing of the program on a larger scale and expansion of the adaptation of the program to include caregivers from more diverse communities with a larger range of sociodemographic experiences, to support all caregivers of pediatric cancer patients.”

Reviewer 2 Report

Comments and Suggestions for Authors

The introduction is prepared correctly, providing background and reference to the problem being addressed. On the other hand, it is lacking to make the purpose of the study clear and explicit. In the last paragraph of the introduction there is a reference to the purpose, but it is not clearly and explicitly formulated. 

The methods are described correctly. 

The authors refer to the recruitment of participants - a detailed description of the inclusion and exclusion criteria is missing here. On what basis was the research sample selected? 

The results are presented correctly, comprehensively. The tables are clear and well described. 

The discussion includes references. 

Clearly separated conclusions and practical implications are missing, please clearly separate this section in the article. 

Author Response

Thank you for taking the time to review this manuscript. Below I have addressed your comments. 

Comment 1: 

The introduction is prepared correctly, providing background and reference to the problem being addressed. On the other hand, it is lacking to make the purpose of the study clear and explicit. In the last paragraph of the introduction there is a reference to the purpose, but it is not clearly and explicitly formulated. 

Response 1: 

Thank you for pointing this out. We emphasized the purpose of the research with a clearer statement on page 2 added to the end of the last paragraph of the introduction. “The purpose of this research was to obtain specific feedback regarding the content and delivery of the CHAT-C program and to pilot test the program for feasibility, acceptability, and preliminary efficacy.”

Comment 2: 

The methods are described correctly. 

Response 2: 

Thank you for this comment.

Comment 3: 

The authors refer to the recruitment of participants - a detailed description of the inclusion and exclusion criteria is missing here. On what basis was the research sample selected? 

Response 3:

Thank you for pointing this out. We selected caregivers of patients who were being treated for cancer at Primary Children’s Hospital. Caregivers had to be adults ages 18 or older. Exclusion criteria was limited to those who did not speak English and access to a video capable device. To clarify that, we have adjusted the wording to be clearer in the first paragraph of the Recruitment & Participants section on page 3. “For both aims of the study, eligible participants were adult (18 years or older) caregivers (typically a parent) of cancer patients who were diagnosed between ages 0 to 25 and treated within the past five years at PCH. Participants also needed to speak English and have access to a smartphone, tablet, or other video-capable device. Participants were recruited by email, text message, and phone call.”

Comment 4:

The results are presented correctly, comprehensively. The tables are clear and well described. 

Response 4:

Thank you for this comment.

Comment 5: 

The discussion includes references.

Response 5: 

We are unaware of any reasoning for excluding references in the Discussion section – we believe these references are necessary to help support the discussion of the results, implications, and conclusions, especially as this is a newer area of research.

Comment 6:

Clearly separated conclusions and practical implications are missing, please clearly separate this section in the article. 

Response 6:

Thank you for this feedback. We added subheadings to the Discussion in order to clearly delineate sections. We also moved around some of the content, which is indicated in yellow highlight. Please see page 12 and 13: 4.1. Implications, 4.2. Limitations, 4.3. Conclusions.”

Round 2

Reviewer 2 Report

Comments and Suggestions for Authors

Thank you very much for all corrections.